# The Influence of Kerosene on Microbiomes of Diverse Soils

**DOI:** 10.3390/life12020221

**Published:** 2022-01-31

**Authors:** Pavel V. Shelyakin, Ivan N. Semenkov, Maria N. Tutukina, Daria D. Nikolaeva, Anna V. Sharapova, Yulia V. Sarana, Sergey A. Lednev, Alexander D. Smolenkov, Mikhail S. Gelfand, Pavel P. Krechetov, Tatiana V. Koroleva

**Affiliations:** 1Institute for Information Transmission Problems (Kharkevich Institute), Russian Academy of Sciences, 127051 Moscow, Russia; f.serval@gmail.com (P.V.S.); m.tutukina@skoltech.ru (M.N.T.); daria.nikolaeva@skoltech.ru (D.D.N.); mikhail.gelfand@gmail.com (M.S.G.); 2Department of Computational Biology, N.I. Vavilov Institute of General Genetics, Russian Academy of Sciences, 119333 Moscow, Russia; 3Faculty of Geography, M.V. Lomonosov Moscow State University, 119991 Moscow, Russia; avsharapova@mail.ru (A.V.S.); sled1988@mail.ru (S.A.L.); krechetov@mail.ru (P.P.K.); korolevat@mail.ru (T.V.K.); 4Center of Molecular and Cellular Biology, Skolkovo Institute of Science and Technology, 121205 Moscow, Russia; Yuliya.Sarana@skoltech.ru; 5Lab of Functional Genomics and Cellular Stress, Institute of Cell Biophysics RAS, 142290 Moscow, Russia; 6Faculty of Chemistry, M.V. Lomonosov Moscow State University, 119991 Moscow, Russia; smolenkov@bk.ru

**Keywords:** soil metagenome, jet-fuel, soil pollution, ecological indicators, controlled study, gasoline, biodegradation, total petroleum hydrocarbons, bearing capacity, xenobiotic compounds

## Abstract

One of the most important challenges for soil science is to determine the limits for the sustainable functioning of contaminated ecosystems. The response of soil microbiomes to kerosene pollution is still poorly understood. Here, we model the impact of kerosene leakage on the composition of the topsoil microbiome in pot and field experiments with different loads of added kerosene (loads up to 100 g/kg; retention time up to 360 days). At four time points we measured kerosene concentration and sequenced variable regions of 16S ribosomal RNA in the microbial communities. Mainly alkaline Dystric Arenosols with low content of available phosphorus and soil organic matter had an increased fraction of Actinobacteriota, Firmicutes, Nitrospirota, Planctomycetota, and, to a lesser extent, Acidobacteriota and Verrucomicobacteriota. In contrast, in highly acidic Fibric Histosols, rich in soil organic matter and available phosphorus, the fraction of Acidobacteriota was higher, while the fraction of Actinobacteriota was lower. Albic Luvisols occupied an intermediate position in terms of both physicochemical properties and microbiome composition. The microbiomes of different soils show similar response to equal kerosene loads. In highly contaminated soils, the proportion of anaerobic bacteria-metabolizing hydrocarbons increased, whereas the proportion of aerobic bacteria decreased. During the field experiment, the soil microbiome recovered much faster than in the pot experiments, possibly due to migration of microorganisms from the polluted area. The microbial community of Fibric Histosols recovered in 6 months after kerosene had been loaded, while microbiomes of Dystric Arenosols and Albic Luvisols did not restore even after a year.

## 1. Introduction

Hydrocarbons serve as the main fuel for transportation engines worldwide, making environmental pollution by hydrocarbons one of the major current ecological threats. Hydrocarbons disrupt intra-soil habitat conditions by filling in the pore spaces and impairing the water and air exchange, thus transforming the composition of soil microorganisms. In case of contamination, most groups die out while the fitter ones resist or even expand. 

Only a few studies have addressed the bacterial communities of terrestrial soils contaminated with different hydrocarbons, such as [1]. Still, they provide insufficient data on the sensitivity and resistance of various groups of soil microorganisms to kerosene contamination in natural conditions. Kerosene is a combustible hydrocarbon liquid derived from petroleum and widely used in industry as a jet-fuel. Particularly, according to the available data, the duration of hydrocarbon biodegradation in different environments may vary from several months to several decades [2,3,4,5,6,7]. Although a wide range of hydrocarbons may adversely affect ecosystems, the research mainstream has focused on the impact of crude and tank oil on aquatic and coastal ecosystems, paying little attention to kerosene-contaminated soils. 

Recent studies have identified bacteria from more than 79 genera capable of degrading petroleum hydrocarbons [8]. In undisturbed soils, the proportion of bacteria capable of decomposing hydrocarbons is negligible [9], while the hydrocarbon contamination results in an increase in their abundance, e.g., *Pseudomonas* and *Burkholderia* in mangroves polluted by tank oil in Okinawa [10]. Bacteria from the *Achromobacter*, *Acinetobacter*, *Alkanindiges*, *Alteromonas*, *Arthrobacter*, *Burkholderia*, *Dietzia*, *Enterobacter*, *Kocuria*, *Marinobacter*, *Mycobacterium*, *Pandoraea*, *Pseudomonas*, *Rhodococcus*, *Staphylococcus*, *Streptobacillus*, and *Streptococcus* genera play vital roles in degrading petroleum hydrocarbons [11]. Interestingly, that some initially rare taxa, e.g., *Alkanindiges* sp., can even become dominant in response to such pollution [11]. 

As alkanes from C5 to C9 are highly volatile, quickly evaporate from the topsoil [12,13], and are not highly toxic to humans, their impact on soil properties and microbiome is understudied. Following the one health approach [14], it is important to understand the danger of certain substances for each component of a whole ecosystem. It is necessary to analyze their impact on the soil microbiota, even if they are non-toxic or little-toxic for humans. 

This study bridges the gap in our knowledge on the environmental consequences of contamination with TS-1 kerosene, which is the fuel most commonly used for commercial aviation in Russia [15]. We investigated three contrast soil reference groups and analyzed the topsoil physicochemical properties and the microbiome composition. We conducted a laboratory pot experiment with the Dystric Arenosols (N 45°43′20″ E 63°11′40″) sampled in Kazakhstan and Albic Luvisols (N 55°11′5″ E 36°25′5″) of Russia and a field experiment with Fibric Histosols (N 55°11′03″ E 36°24′58″) and the same Albic Luvisols in Russia.

Our research objective was to characterize the responses of soil microbiomes to kerosene contamination under humid and semi-arid climates. We observed that the kerosene content decreased faster under natural conditions than in a laboratory experiment. The response of soil microbiomes to kerosene contamination was similar in the laboratory and field experiments, that is, in the steady and natural environments, respectively. 

## 2. Materials and Methods

The selected soils are representative of humid landscapes of temperate mixed forests (the Kaluga region in the Russian Federation) and semi-arid landscapes of deserts (the Kyzylorda region in the Republic of Kazakhstan). 

The Kaluga region is located at the south-east of the Smolensk–Moscow Upland. The region is characterized by a snowy fully humid climate with a warm summer [16] and with an annual precipitation of about 650 mm. The growing season lasts from May to September. The parent rocks of well-drained interfluves and slopes are the Quaternary loess-like loams at least 2 m thick underlain by lacustrine sediments. Fibric Histosols with a peat thickness of about 1.5 m are formed in local depressions of the interfluves, on the clayic or loamic rocks under the sphagnum mosses. Albic Luvisols represent the predominant soil in this region. 

The Baikonur Cosmodrome area encompasses a part of the Daryalyk residual plateau represented by an undulating plain composed of stony clay loams and sandy loams, and a part of a terraced valley of the Syr Darya river [17]. Arenosols, Calcisols, and Gypsisols are the most typical soils of the region. The region is characterized by a cold arid desert climate [16], with the annual precipitation of about 100–120 mm. The most favorable hydrothermal regime for the biota development takes place in the periods from March to June and from September to November [7,17]. 

### 2.1. Pot Experiment

Two A-horizon soil samples (20 kg, natural moisture) were sieved using sieves with a 3 mm mesh, cleaned of roots and other coarse fractions, and dried to the air-dry state. Before the pot experiment, the samples were moistened to a level of 60% of the maximum field capacity. To reach the uniform absorption of distilled water, the soil was thoroughly mixed each time after water was added. The soil moisture before the kerosene contamination, determined gravimetrically, was 22.7 ± 0.5% and 5.2 ± 0.3% for Albic Luvisols and Dystric Arenosols, respectively. Then, the samples were stored in plastic bags (temperature 18–22 °C) for three days to activate the soil microbiome and were carefully and periodically stirred to homogenize while preserving soil micro-aggregates [18]. 

After three days (time point 0), both soil samples were divided into six subsamples (Appendix A). One subsample without any contamination was used as a control. The remaining five subsamples were treated with various loads (1, 5, 10, 25, and 100 g/kg of soil, separately) of kerosene. Low kerosene loads (1, 5, and 10 g/kg) were applied as a spray. High loads (25 and 100 g/kg) were applied from a watering can. The loads were selected based on the previous results on the response of vegetation [18], cultivated soil microorganisms [19], and cellulolytic bacteria [20] to kerosene contamination. 

All subsamples (12 in total) were placed into the glass containers with hermetically sealed iron lids to the bulk density of 1.47 ± 0.04 and 0.92 ± 0.09 kg/dm^3^, which is typical of natural Dystric Arenosols and Albic Luvisols, respectively. The experiment lasted for 12 months in 2019–2020 at a temperature of 18–22 °C. Every 5 days, the containers were opened for ventilation and, if needed, moisturized with distilled water.

### 2.2. Field Experiment

The field experiment was conducted on Albic Luvisols under a spruce–aspen forest and Fibric Histosols under a subshrub–sphagnum pine forest in the Kaluga region in 2020–2021. 

Experimental plots with a size of 50 × 50 cm (Appendix A) were selected based on the microtopography, comprising spatial homogeneous microsites without visible microslopes of the soil surface. The surface of each plot was cleared of plant litter to reduce possible redistribution of kerosene over the soil surface and to achieve better absorption into the topsoil. Plots were contaminated with the same kerosene TS-1 loads for the 0–10 cm topsoil layer as in the pot experiment.

The field experiment lasted for 12 months from 2020–2021 under the natural conditions (Appendix A). Topsoil samples were collected in summer (3 and 360 days after treatment), fall (90 days), and early winter (180 days).

### 2.3. Soil Sampling and Chemical Analysis

In total, 288 topsoil (0–10 cm) samples, 50 g each, were collected, in triplicate, 3, 90, 180, and 360 days after the kerosene contamination. For the chemical analysis, each replicate was placed into a glass jar with metal lids. Then, subsamples of 1–2 g were collected for the isolation of total DNA. 

In 96 soil samples (1 mixed sample of the triplicate replicas), chemical analysis was performed immediately after soil sampling using routine techniques for pH, moisture, cation exchange capacity (CEC), content of soil organic matter (SOM), available phosphorus (Pav) and potassium (Kav), exchangeable ammonium (NH_4_^+^), and water-soluble nitrate (NO_3_^−^), as described in Appendix A.

In 288 soil samples, kerosene concentration was determined using the original method partially reported in [21]. Briefly, we used a system of the Agilent 7890 V gas chromatograph by Agilent Technologies (Santa Clara, CA, USA) equipped with a 5977 A quadrupole mass-spectrometric detector. Samples of natural moisture weighing 1–2 g were placed into glass flasks. After that, 0.2 cm^3^ of 1 g/dm^3^ 1-chlorooctadecane solution (internal standard), 2 g of Na_2_SO_4_, and 10–20 cm^3^ of dichloromethane were added, and the flasks were loosely closed. The extraction was carried out for 15 min in an ultrasonic bath. The extract was filtered through a paper filter (a red ribbon), previously washed with 3 cm^3^ of dichloromethane. The flask and the filter were rinsed with 5 cm^3^ of dichloromethane. The filtrates were combined and transferred into 2 cm^3^ glass vials. Two parallel measurements were made. To calibrate the chromatograph, we used the soil samples unpolluted with kerosene and jet-fuel used for experiments. The retention time range of the components of kerosene and the retention time of the internal standard were determined using the obtained chromatograms. We found the total area of all peaks on the chromatogram using the total ion current and the area of the internal standard and calculated the ratio of the total peak area of all kerosene components to the peak area of the internal standard.

### 2.4. DNA Extraction, Amplification and Sequencing

Total DNA from soil subsamples was isolated using DNeasy PowerSoil (Qiagen, Hilden, Germany). Three independent samples were collected from each pot and plot. Further, they were placed immediately either into a PowerBead tube, or into a sterile 1.5 mL tube and stored at −20 °C prior to extraction. Each sample of 300 mg was placed into a PowerBead tube, then 60 μL of the C1 buffer were added, and the tube was inverted 4–5 times to mix the reagents. The samples were then disrupted using TissueLyser II or TissueLyser LT (Qiagen, Hilden, Germany, 10 min, 30 Hz). Further DNA purification was made according to the manufacturer’s protocol.

DNA concentration was measured on the Qubit 1 fluorimeter using the Qubit DNA HS kit (Thermo Fisher Scientific, Waltham, MA, USA) and ranged from 2 to 3 ng/μL. Variable 16S rRNA regions were amplified with two primer combinations, V3/V4 and V4/V5 (341F 5′–CCTAYGGGRBGCASCAG–3′ and 806R 5′–GGACTACNNGGGTATCTAAT–3′; 515F 5′–GTGCCAGCMGCCGCGGTAA–3′ and 907R 5′–CCGTCAATTCCTTTGAGTTT–3′, respectively) using the Phusion polymerase (New England Biolabs, Ipswich, MA, USA) and the following program: 

Step 1:   95 °C   2 min (initial DNA melting);

24 cycles as follows: 

Step 2:   95 °C   30 s (melting);

Step 3:   58 °C   30 s (primer annealing);

Step 4:   72 °C   40 s (synthesis);

Step 5:   72 °C   5 min.

To prepare sequencing libraries, amplicons were purified using the AMPure XP beads (Beckman Coulter, Brea, CA, USA) according to the manufacturer’s protocol. The amplicon concentrations were measured on the Qubit 1 fluorimeter using the Qubit DNA HS kit (Thermo Fisher Scientific, Waltham, MA, USA) and ranged from 0.8 to 40 ng/μL. Samples containing no DNA processed in the same laboratory were used as negative controls, their concentrations were in the range of 0.2–0.6 ng/μL.

To skip the adaptor ligation step, all primers already contained Illumina 1 (forward primer) or Illumina 2 (reverse primer) adaptors. Index PCR was made using the Phusion polymerase (New England Biolabs, Waltham, MA, USA) and the Nextera XT Index kit (Illumina, San Diego, CA, USA) following the manufacturer’s protocols. The library concentrations were measured on the Qubit 1 fluorimeter using the Qubit DNA HS kit (Thermo Fisher Scientific, Ipswich, MA, USA), and the libraries were sequenced on Illumina MiSeq with the read length of 250 bp (MiSeq Reagent Kit v2). The average number of reads was 75,000 per sample.

### 2.5. 16S rRNA Sequencing Data Analysis

The quality of reads was analyzed with FastQC [22]. Quality filtering, denoising, paired reads merging, and chimera filtering were performed using the R package DADA2 v. 1.14.1 [23]. Parameters for DADA2 were modified to lose less than 50% of reads in the pipeline (maxEE = c(3,3), minOverlap = 8, maxMismatch = 1, minFoldParentOverAbundance = 8). The obtained Amplicon Sequence Variants (ASV) tables were analyzed and filtered of potential contaminants using the R package phyloseq [24] and decontam [25], respectively, taking into account the amplicon concentrations and ASVs found in negative controls. Low total numbers of reads in negative controls and the results of the decontam analysis indicated low levels of contamination. After filtration and removing the singleton ASVs, the mean number of reads in the samples was approximately 52,000. 

Taxonomic labels were assigned to ASVs using IdTaxa [26] from the R package decipher [27] trained on 16S rRNA gene sequences from the SILVA database [28]. After that, ASVs assigned to “Chloroplast” and ASVs not classified at the domain level were filtered out and all samples were rarefied to a standard number of reads (10,000 reads) in order to account for differences in sequencing depth. 

Multiple alignment of ASVs and construction of phylogenetic trees was conducted using AlignSeqs [29] from the decipher package and FastTree v.2.1.11 [30], respectively. 

The metabolic potential of microbial communities in the samples was predicted with Picrust2 [31]. Principal Component Analysis (PCA) (function prcomp in R) on the relative abundance of predicted MetaCyc pathways was used to visualize metabolic potential similarities between the samples. 

### 2.6. Assessment of Soil Microbial Community Diversity and Statistical Analysis

The diversity analysis was carried out using the R package vegan [32]. The within-sample (alpha) diversity was estimated using the Shannon index. The between-samples (beta) diversities were estimated using the Bray–Curtis dissimilarity and weighted Unifrac. To visualize the between-sample diversity, dimensionality reduction with the Principal Coordinate Analysis (PCoA) was performed. 

To ensure that changes in bacterial communities did not result only from their death under high kerosene concentrations, the relative amount of bacteria in soil samples with or without kerosene load at different time points was measured by quantitative PCR with 341F–806R primers. qPCR was performed using qPCRMix HS-SYBR (Evrogen, Moscow, Russia) and DT-Lite machine (DNA Technology, Moscow, Russia). Each sample was assayed in triplicate. The differences were calculated using ΔCt. 

Permutational Multivariate Analysis of Variance (PERMANOVA) using distance matrices implemented in the adonis function from the vegan package was used to estimate the significance of microbiome differences between the studied conditions. The FDR correction (Benjamini-Hochberg) for multiple testing was applied separately for different tests in different soils. Aldex2 [33] was used to search for bacterial groups which proportion significantly differed between the studied conditions. 

## 3. Results and Discussion

The chosen soils are most representative for studying kerosene contamination. The soils of the Kaluga region could be considered as the background for the Moscow region [20]. The latter’s contamination with kerosene occurs because it is the largest aviation hub in Russia. Dystric Arenosols are the most vulnerable soils in the Baikonur Cosmodrome area, where ‘Soyuz’ vehicles propelled by kerosene are launched and an airport is situated [7,15].

Since the choice of a 16S rRNA region for sequencing may introduce a bias to the final results [34], we tested two different sets of primers for 16S rRNA region V3V4 and V4V5. The results obtained with these two 16S rRNA regions were highly consistent (Spearman correlation on bacterial family abundance greater than 0.75), while V3V4 resolved more microbial families (Appendix A). For example, Pseudomonadaceae, Chthoniobacteraceae, and Moraxellaceae, highly abundant in several samples, were observed with V3V4 but almost not detected with V4V5. The estimated alpha-diversity was also generally higher in case of the V3V4 analysis (Figure 1). An exception was represented by the Dystric Arenosols samples, where several bacterial families, i.e., Alcaligenaceae and Nocardiaceae were relatively more abundant when V4V5 was used. For simplicity, in the main text, the V3V4 data are used for plots, while the plots for V4V5 are provided in the Appendix A. These discrepancies are minor and do not influence the conclusions.

The microbiomes of the topsoils of different properties (Dystric Arenosols, Fibric Histosols, and Albic Luvisols) showed similar responses to equal kerosene loads. In most of the contaminated soils, the proportion of anaerobic bacteria-metabolizing hydrocarbons was elevated, whereas the proportion of aerobic bacteria was reduced. During the field experiment, the soil microbiome recovered faster than in the pot experiments, possibly due to migration of organisms from the surrounding uncontaminated environment.

### 3.1. Soil Similarities and Differences in Physicochemical Properties and Microbiome Composition

Prior to kerosene pollution, all soil communities under consideration were dominated by Proteobacteria, Actinobacteriota, Acidobacteriota, Verrucomicrobiota, and Bacteroidota. These phyla comprised nearly 80% of all bacteria in communities (Figure 2C and Figure 3A), consistent with the previous results on soil microbiomes [35,36], in particular on Aridisols of (semi)arid regions [37,38], Albic Luvisols [39,40,41,42], and Fibric Histosols [43,44,45,46] of humid environments. Dystric Arenosols with the alkaline environment and the minimal available phosphorus and SOM (Figure 2A) had an increased fraction of Actinobacteriota, Firmicutes, Nitrospirota, Planctomycetota and less Acidobacteriota, Verrucomicobacteriota (Figure 2C). In highly acidic Fibric Histosols with the maximal content of SOM and available phosphorus (Figure 2A), the fraction of Acidobacteriota was higher, while the fraction of Actinobacteriota was lower (Figure 2C). Albic Luvisols occupied an intermediate position in terms of both physicochemical properties and the microbiome composition (Figure 2).

In the uncontaminated samples, the changes in microbial communities with time (Appendix A) were less pronounced compared to the changes in the contaminated soils, indicating high stability of the studied soil microbiomes (Figure 2C and Figure 3).

Based on metabolic properties estimated with Picrust2, the Dystric Arenosols samples formed a clearly separated and compact cluster (Appendix A). This separation resulted from the specific low-level pathways within such categories rather than changes in high-level pathway categories. 

### 3.2. Temporal Changes in the Kerosene Concentration and the Physicochemical Soil Properties

As a null hypothesis of the field and laboratory experiments, we assumed monotonous changes in the analyzed soil properties (pH value, soil moisture, CEC, content of SOM, available P and K, exchangeable ammonium, water-soluble nitrates) dependent on the kerosene load and smoothing of the observed differences over time due to a decline in kerosene concentration and self-recovery processes in soils.

Indeed, the content of kerosene continuously decreased, with a more rapid and pronounced decline in the field experiment (Figure 4). Even only 3 days after pollution of Albic Luvisols, the observed kerosene concentration under the same loads was 1.5–4-fold higher in the pot experiment than in the field experiment (Appendix A), implying a more intensive soil self-restoration of volatile hydrocarbons in natural conditions. After 90 and 180 days, the concentration dropped by up to several orders of magnitude from the initial level in all samples from the field experiment, whereas in the pot experiment, it showed a slow decline in moderately contaminated samples and only a slight decrease in highly contaminated ones. After a year, in the samples with initial loads of 10 g/kg or less, kerosene was detected in negligible concentrations (<0.1 g/kg) in all soils. In the samples with higher initial loads, kerosene was almost not detectable in Fibric Histosols and reached approximately 0.1–0.3 g/kg (initial load of 25 g/kg) and 0.3–1.1 g/kg (initial load of 100 g/kg) in the Albic Luvisols samples from the field experiment. The samples of the laboratory pot experiment with high initial kerosene loads maintained higher kerosene concentrations. 

In all experiments and soils, the controlled physicochemical properties did not show similar changes after kerosene contamination, e.g., the SOM content and moisture increased during the pot experiments (3, 90, and 180 days after treatment) and did not change during the field experiments. It could have resulted from a thorough imbibition of soil with kerosene in a lab, which cannot be achieved in a field due to the absence of soil agitation and prevention of kerosene contamination in wet soil microcompartments. A water film on the surface of soil aggregates prevents them from imbibition of nonpolar hydrocarbons and being affected by kerosene [47,48]. 

In the pot experiment with Dystric Arenosols, a higher kerosene load was associated with lower pH (by 0.2–0.3 units; Appendix A). Initially (3 days after treatment), the SOM content increased more than 5-fold from samples with the minimal kerosene load to samples with the maximal one, reaching 0.3–0.6% by the end of the year. The soil moisture behaved similarly: in the most-contaminated sample, the moisture was twice as high as that in the uncontaminated samples (3, 90, and 180 days after treatment) and varied stochastically after a year. Monotonous dependencies on the kerosene load were not observed for CEC, available P and K, water-soluble nitrates, and exchangeable ammonium. 

In the pot Albic Luvisols samples, the SOM content, moisture, and CEC grew at increasing kerosene load and then with time returned to the initial values. In all time points there were no monotonous dependencies on the initial kerosene load for pH, available K and P, water-soluble nitrates, and exchangeable ammonium. After a year, no monotonous dependency on the initial kerosene load was observed for any measured parameter. 

In the field Albic Luvisols experiment, the content of available K (during the whole observation period) and exchangeable ammonium (days 3 and 180) increased with the kerosene load, whereas there was no monotonicity for pH, SOM, available P, water-soluble nitrates, moisture, and CEC. 

In Fibric Histosols, no monotonous trends relative to the kerosene load were identified for any of the analyzed parameters. Therefore, the physicochemical properties of this soil tend to be the most susceptible to kerosene influence. 

Previously, it was reported that the kerosene pollution yielded an increase in the concentration of organic carbon and a decrease of available nitrogen and phosphorus [49,50,51]. Injection of kerosene only weakly influenced CEC and the exchangeable acidity of soils [51], although the soil pH shifted towards neutral values [50,51]. 

### 3.3. Taxonomic Composition of Soil Samples Treated with Kerosene

In untreated Albic Luvisols and Fibric Histosols from the field experiment, only weak seasonal variability was observed in the microbiome composition (Figure 2C and Figure 3). 

The treatment of soil samples with kerosene led to significant changes in the soil microbial composition and diversity (Appendix A, Figure 1), which strongly depended on the kerosene load, forming three distinct patterns for slightly (0 and 1 g/kg), moderately (5 and 10 g/kg), and heavily (25 and 100 g/kg) contaminated samples. 

The alpha-diversity of slightly contaminated samples stayed almost stable for all soils and dramatically decreased in heavily contaminated samples (Figure 1). In moderately contaminated samples, the alpha-diversity decreased until 180 days and then started to increase towards the initial levels for some of the samples, while continuing to decrease for others. The Fibric Histosol samples were an exception, since even in heavily contaminated samples the alpha-diversity started to restore after 180 days. In the pot experiments we observed an increase of the alpha-diversity throughout 90 days after treatment, possibly arising from the ongoing adaptation of soil microbial communities to the laboratory conditions. We also observed a decrease of diversity by 360 days even in the control samples possibly due to the exhaustion of nutrients and the lack of migration opportunity in a closed system. 

To verify that the observed changes did not stem solely from the eliminating sensitive phyla, we performed qPCR for some of the samples (Appendix A); this showed that in the field experiment both with Fibric Histosols and Albic Luvisols, the total abundance of 16S rRNA amplicons did not change with time both in the control and highly contaminated samples, compared to the abundance in control samples at day 3. However, in the highly contaminated samples of Albic Luvisols (pot experiment), the abundance of amplicons dramatically decreased with time, while being stable in the control. While the 16S rRNA sequencing demonstrated the similarities of the microbiome response in the pot and field experiments with all treated soils, particularly, an increase in the relative abundance of kerosene-degrading bacteria, the absolute bacterial abundance did not reach the level observed before contamination, indicating that the contamination in the pot conditions was much more stressful for the microbiome than in the field experiment. A natural caveat is that limitations of the qPCR analysis, e.g., different copy number of the 16S rRNA loci or different efficiency of PCR for bacterial taxa could have impacted the observations. 

In PCoA plots with the weighted UNIFRAC serving as metric, the first axis (explaining 37–63% of the variance) always reflected the response to the kerosene contamination. In all studied systems, slightly and heavily contaminated samples could be easily separated when projected onto this axis (Figure 5). The functional role of the second axis was less prominent, but in several cases, it could be connected with the period after treatment. Therefore the kerosene contamination appears to be the main factor introducing variance between the samples. PCoA with the Bray–Curtis dissimilarity was more sensitive to batch effects, separating samples sequenced in different runs (plots B and D in the Appendix A). In the pot experiment, samples were sequenced in three runs: (i) samples collected on day 3; (ii) samples collected on days 90 and 180; (iii) samples collected on day 360. Using Bray–Curtis dissimilarity for estimate beta-diversity between the samples from these three runs resulted in clearly distinct clusters along the Axis 1 on PCoA plots (Appendix A), while use of weighted UNIFRAC prevented such clustering. For instance, control samples from different days were closer to each other rather than to highly contaminated samples from the same run. All samples from the field experiment were sequenced in one run and for them we did not observe any drastic difference between PCoA plots based on Bray–Curtis dissimilarity and weighted UNIFRAC (Appendix A).

In the pot experiment, the samples collected on the third day after kerosene treatment formed a dense cluster with smaller mean beta-diversity between samples with different kerosene loads, as compared to the field experiment (Figure 5 and Appendix A). This discrepancy might result from several causes. All pot microbiomes could be highly and similarly stressed by the soil preparation procedure and environmental change, while the DNA of bacteria that died immediately following the kerosene pollution could be preserved longer as the surviving bacteria were suppressed and inactive. On the other hand, in the field experiments, the bacterial community could react actively to the kerosene treatment as some soil microcompartments might be uncontaminated and bacteria could migrate naturally. However, these differences were relatively small and statistically significant in the pot experiment and insignificant in the field one. The cause might have lain in the small number of samples and a high diversity between those from the same group in the field experiment (Appendix A).

Compared to the control samples (zero kerosene pollution, day > 3), the highly contaminated samples (kerosene loads of 25 and 100 g/kg, day > 3) demonstrated that the fractions of Burkholderiaceae and Sphingomonadaceae increased in response to the kerosene contamination at least at some time points in all studied soils. Moreover, the fractions of *Acetobacteraceae*, *Caulobacteraceae*, *Mycobacteriaceae*, *Nocardiaceae*, *Oxalobacteraceae*, *Pseudomonadaceae*, and *Solimonadaceae*, as well as unclassified families of the *Burkholderiales* order were elevated in three out of four studied systems (Figure 3b, Appendix A). 

At the phylum level, the most remarkable changes at high kerosene loads were the increase of the relative abundance of Proteobacteria in all soils and a decrease of Acidobacteriota and Actinobacteriota (Figure 3A and Appendix A). Given that, the increased relative abundance of Proteobacteria suggested that the observed changes were not caused by survival of spore-forming bacteria as opposed to extinction of most other bacteria, since Proteobacteria do not form endospores [52]. 

In all soils except Fibric Histosols, upon the high kerosene load, we observed several dominant bacterial families that expanded up to 50% of the total bacterial community (Figure 3B). For example, in Albic Luvisols in the pot experiment, *Burkholderiaceae* and *Yersiniaceae* became dominant families in the moderately and highly contaminated samples, reaching, respectively, up to 56% and 22% of the total community in some samples. Such relative increase of abundance was not soil-specific, because in the field experiment, other bacterial families became dominant in the same Albic Luvisols soil; for example, in several highly contaminated samples, *Rhodocyclaceae* constituted 38% of the total community. *Caulobacteraceae* and *Pseudomonadaceae* increased in almost all soils and represented the dominant family in several samples of two different soils, field Albic Luvisols (*Caulobacteraceae* up to 10%, *Pseudomonadaceae* up to 15%) and pot Dystric Arenosols (*Caulobacteraceae* up to 12%, *Pseudomonadaceae* 22%). Similarly, in Dystric Arenosols, the families that were enriched in almost all soils, such as *Sphingomonadaceae* (up to 21%) and *Nocardiaceae* (up to 16%), were predominant in the moderately and highly contaminated samples.

Most bacterial families that either increased their relative abundance in most of the studied soils, or became dominant in some samples, belonged to Proteobacteria. An increase in Proteobacteria is a frequent response to soil contamination by crude oil or its derivates [53,54,55,56]. Some families belonged to the phylum Actinobacteria, also known to respond to contamination [53]. Bacterial families that were expanded in moderately and highly contaminated soils comprise representatives with a known ability to survive and degrade aliphatic or/and aromatic hydrocarbons. For instance, *Burkholderiaceae* and *Sphingomonadales* are well-known for their ability to degrade polycyclic aromatic hydrocarbons [57]. Some *Sphingomonadales* can degrade petroleum hydrocarbons including polycyclic aromatic hydrocarbons [58]. *Burkholderiaceae* are enriched in many oil-contaminated soils [53,59], whereas *Pseudomonadaceae* dominate bacterial communities of oil-contaminated mangrove sediments [10]. Members of *Pseudomonadaceae* are among the main bacteria responding to biodiesel contamination [56], degrade diesel in Antarctic soils [60] and are dominant degraders of polycyclic aromatic hydrocarbons in Arctic soils [61]. *Caulobacteraceae* are highly enriched in oil contaminated soils [53,62], and *Solimonadaceae* have been extracted from oil contaminated water samples [63]. Representatives of *Rhodocyclaceae* degrade aromatics compounds both in aerobic and anaerobic conditions [64]. *Oxalobacteraceae*, while being members of uncontaminated soil communities [53,62], also act as key functional n-alkene degraders in crude-oil contaminated soils [65]. Several members of *Acetobacteraceae* degrade hydrocarbons in slightly acidic environments [66].

Members of phylum Actinobacteria, e.g., *Nocardioidaceae*, can degrade alkane and aromatic compounds in contaminated soils [67], *Mycobacteriaceae* are propagated in soils contaminated with petroleum and diesel [68]. 

The metabolic potential determined with Picrust2 clearly differentiates slightly contaminated (kerosene load is less than 1 g/kg) and moderately and highly contaminated (5–100 g/kg) samples (Figure 6 and Appendix A). On the PCA plot, the moderately and highly contaminated samples were shifted in the same direction as compared to the slightly contaminated samples in all soils. This could indicate that in response to kerosene contamination, abundance of similar metabolic pathways decreased or increased. Interestingly, such a shift appeared in the samples from the pot experiment starting from day 90, while the samples from the field experiment responded faster and showed the shift from day 3. The Aldex2 analysis of MetaCyc pathways differentially abundant in slightly and highly contaminated samples showed that the highly contaminated samples featured a larger number of degradative pathways, specifically, the degradation of aromatic compounds, amino acids, and secondary metabolites (Appendix A). This could result from an increase in the relative abundance of bacteria capable of degrading aromatic hydrocarbons from kerosene. Slightly contaminated samples featured more pathways related to the biosynthesis of cofactors, nucleotides, amino-acids, lipids, carbohydrates, secondary metabolites, and aromatic compounds; still, several pathways of polymer degradation and respiration were enriched. 

Furthermore, the soils of the same geographic origin demonstrated stronger similarity to each other in the fraction of metabolic pathways. Fibric Histosols and Albic Luvisols from the humid environments of the Kaluga region being the most similar, and the soils of semi-arid environments in Kazakhstan were significantly different from the Kaluga region soils. 

While the studied soils had quite different initial microbial communities, several groups of bacteria simultaneously decreased the relative abundance in response to the kerosene contamination in almost all soils (Appendix A); these were unclassified *Acidobacteriae*, *Bryobacteraceae*, *Haliangiaceae*, *Nitrosomonadaceae*, *Pedosphaeraceae*, *Pyrinomonadaceae*, *Solibacteraceae*, unclassified Kapabacteriales, Polyangia, and RCP2-54, as well as the WD2101 soil group. The resolution at genus level is shown in Appendix A.

#### 3.3.1. Albic Luvisols, the Pot Experiment

In the PCoA plot with weighted UNIFRAC (Figure 5), all samples were arranged across two axes that visually correlated well with the experimental conditions. All samples from the time point 0 (day 3) clustered together in one corner (upper left) of the plot (*p*-value < 0.0001 for the V3V4 data on; however, this was insignificant for the V4V5 data, Appendix A). 

With time, the moderately and highly contaminated samples were shifted along Axis 1 (explaining 63% of variation), whereas the slightly contaminated samples were shifted along Axis 2 (approximately 16% of variation). Therefore, Axis 1 may reflect the response to kerosene treatment, while Axis 2 reflects the influence of laboratory conditions. Given that, the samples with high kerosene loads tended to move faster along the Axis 1, while the samples with intermediate loads showed some recovery potential, as after 1 year of incubations they were more similar to the controls than after 6 months. The main bacterial families that increased during this shift along the Axis 1 were Yersiniaceae, Burkholderiaceae, and Solimonadaceae; bacterial families that considerably changed in relative abundance in the contaminated samples are shown in Figure 7 and Appendix A.

#### 3.3.2. Dystric Arenosols, the Pot Experiment

In the PCoA plot, samples again arrange across Axis 1, according to the added kerosene load, with more contaminated samples located farther from the control samples (Figure 5). However, by day 360, the samples with the lowest kerosene load (1 g/kg) and moderately contaminated samples appear to move back, somewhat closer to the control samples, unlike the highly contaminated ones. This might result from the reconstitution of the soil communities. In moderately contaminated samples, this effect was seen in the taxonomic barplot, as in Proteobacteria, the relative abundance of which had increased in the initial response to kerosene and declined after 1 year of incubation (Figure 3A). Bacterial families that considerably changed in abundance in the contaminated samples are shown in Figure 8, Appendix A.

#### 3.3.3. Albic Luvisols, the Field Experiment

Microbial communities in the field experiments were expected to exhibit a noisier behavior compared to the pot experiment due to environmental seasonal changes, and a faster recovery due to kerosene removal by natural degradation (oxidation, illuviation, and evaporation) and possible colonization by bacteria from adjacent, untreated areas. Despite soil sampling in summer, autumn, and winter, the alpha-diversity in slightly contaminated samples showed an elevated temporal stability (Figure 1C), while in highly contaminated samples, it decreased after the treatment and was not restored to the initial state even 1 year after the treatment, continuing to decrease. Hence, 1 year after treatment in the field experiment, the microbial communities in highly contaminated soils were still stressed in spite of the low concentration of kerosene in all samples. In moderately contaminated samples, the alpha-diversity initially declined, but 1 year later, it could become even higher than at the initial time point, as in the series with the initial kerosene load 10 g/kg and in one of the two series with kerosene load 5 g/kg (the other one showing a slight decrease of the alpha-diversity). This could be interpreted as microbiome recovery. 

In PCoA, the difference between samples can be observed already on day 3, with highly contaminated samples being located separately from others (*p*-value < 0.0001, Appendix A, Figure 5). Similar to Dystric Arenosols, the slightly contaminated samples formed a tight cluster in one half of the plot, while the moderately and highly contaminated samples were shifted along the Axis 1, the shift being more pronounced for the highly contaminated samples. 

On the taxonomic barplot, the most of moderately contaminated samples showed a tendency to reconstitute, with bacterial communities becoming similar to the control after 1 year (Figure 3B). The changes in bacterial communities in the highly contaminated samples became more pronounced over the whole experiment. Caulobacteraceae and Pseudomonadaceae expanded after 90 and 180 days of incubation, and Rhodocylaceae and Moraxellaceae propagated after 180 and 360 days (Figure 9 and Appendix A).

#### 3.3.4. Fibric Histosols, the Field Experiment

Soil communities in Fibric Histosols showed the most pronounced ability to restore the alpha-diversity after kerosene contamination (Figure 1). This might result from fast soil recovery from kerosene, its level being not detectable in most samples after 6 months. In the moderately and highly contaminated samples, the alpha-diversity was the lowest after 90 days, but was at least partially restored by day 180. After 360 days, the alpha-diversity was similar to the initial level in most samples of the series. 

The differences in average beta-diversity were insignificant both between groups of samples with different kerosene loads at the same day, or between groups with the same kerosene load on different timepoints (Appendix A). This is most likely caused by the high heterogeneity of the Fibric Histosols samples. However, the moderately and highly contaminated samples were shifted compared to the slightly contaminated samples along Axis 1 of the PCoA plot (Figure 5). Interestingly, this shift decreased with time in most samples. Addition of moderate and high amounts of kerosene reduced the fraction of Acidobacteriota and increased the fraction of Proteobacteria. These changes were already observed by day 3 (Figure 3A). Burkholderiaceae, Moraxellaceae, Mycobacteriaceae, Chthoniobacteraceae, and unclassified Vampirivibrionia expanded in relative abundance from day 3 to day 180 (Figure 3B, Figure 10 and Appendix A), but then the fraction of these families decreased again. Hence, while the alpha-diversity of communities was almost restored after 180 days, their taxonomic composition was different compared to the slightly contaminated samples (Appendix A).

## 4. Conclusions

Our results demonstrate that kerosene cannot be considered as a safe substance for soil bacteria, having a detrimental effect on soil microbiome. Even after its clearance, the composition of microbiome (especially at high loads) can drastically differ from the initial one, which we call a ‘kerosene label’. Kerosene pollution should be taken into account when conducting ecological soil monitoring and for agricultural purposes. Moreover, the obtained results may guide the selection of candidate bacteria for bioremediation and enable ranging soils according to their resistance or vulnerability to hydrocarbon contamination. 

We show that kerosene disappears faster in the natural conditions of a field experiment as compared to a laboratory experiment with limited aeration, drainage, and the migration of bacteria and substances. One year after the treatment all studied soils contained no more than 1.4 g/kg of kerosene upon a high initial kerosene load (>25 g/kg) and were kerosene-free at lower initial loads. Consistent changes in the physicochemical properties of soils were observed in even shorter periods and only in the pot experiment. 

The response of the studied soil systems to adverse kerosene impact were somewhat similar, as the proportion of bacteria resistant to hydrocarbons and/or able to metabolize hydrocarbons was increased. After kerosene input, the alpha-diversity of all moderately and highly contaminated soils decreased due to extinction of many minor bacterial groups and the development of dominant bacteria taxa comprising the degraders of aliphatic and aromatic hydrocarbons. Despite the fact that kerosene was gradually dissipated and the soil physicochemical properties generally restored to the initial levels, the microbiome composition in moderately and highly contaminated samples recovered slowly, and not in all studied systems. In the highly contaminated soils (25–100 g/kg), the changes were irreversible within 1 year in Dystric Arenosols and Albic Luvisols, but reversible in Fibric Histosols. The microbocenoesis of the moderately contaminated samples (5–10 g/kg of soil) of Albic Luvisols restored faster during the field experiment and remained disturbed in the pot experiment. Adding 1 g/kg of kerosene only slightly changed the composition of all studied soil microbiomes. During the pot experiment, the alpha-diversity of the soil microbial community continued to decline after 1 year even in the control samples. 

We conclude with the following useful practical suggestions. (1) Weak seasonal variation of the microbiome composition of untreated soils in the field experiments from Summer to Fall and Winter allows for the comparative analysis of soil bacterial communities sampled in different seasons. (2) As the data obtained with two 16S rRNA regions were consistent, and more bacterial families could be identified using the V3V4 region compared with the V4V5 region, the former is more suitable for the characterization of soil microbiomes. (3) Weighted UNIFRAC accounting for the phylogenetic relationships of the soil bacteria is more robust to sequencing errors than the Bray–Curtis dissimilarity measure. 

## Figures and Tables

**Figure 1 life-12-00221-f001:**
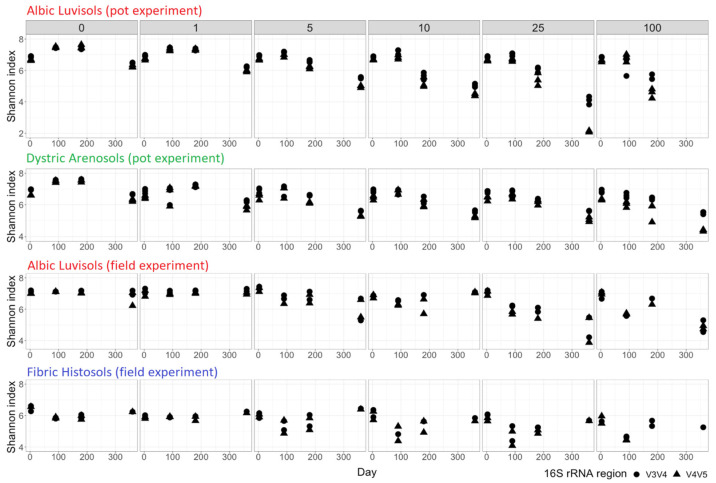
Alpha-diversity estimated by the Shannon index at the ASV level. The numbers above the histograms represent the initial kerosene loads, in g/kg. 16S rRNA regions are shown by the dot (V3V4) and triangle (V4V5).

**Figure 2 life-12-00221-f002:**
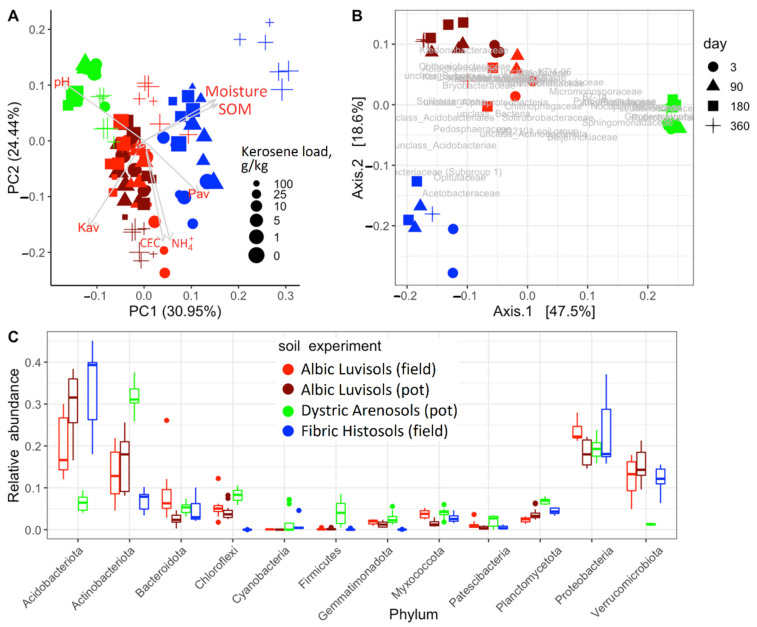
Differentiation of the studied soils in the pot and field experiments: (**A**) principal component analysis (PCA) of the soil chemical properties; (**B**) principal coordinate analysis (PCoA) of the microbiome composition based on weighted UNIFRAC as a metric; (**C**) the relative abundance of the top 12 most frequent phyla among all uncontaminated samples in all experiments (all time points). The soil is shown by the color: (dark) red, Albic Luvisols; green, Dystric Arenosols; blue, Fibric Histosols. The microbial composition was estimated using the V3V4 region of 16S rRNA gene.

**Figure 3 life-12-00221-f003:**
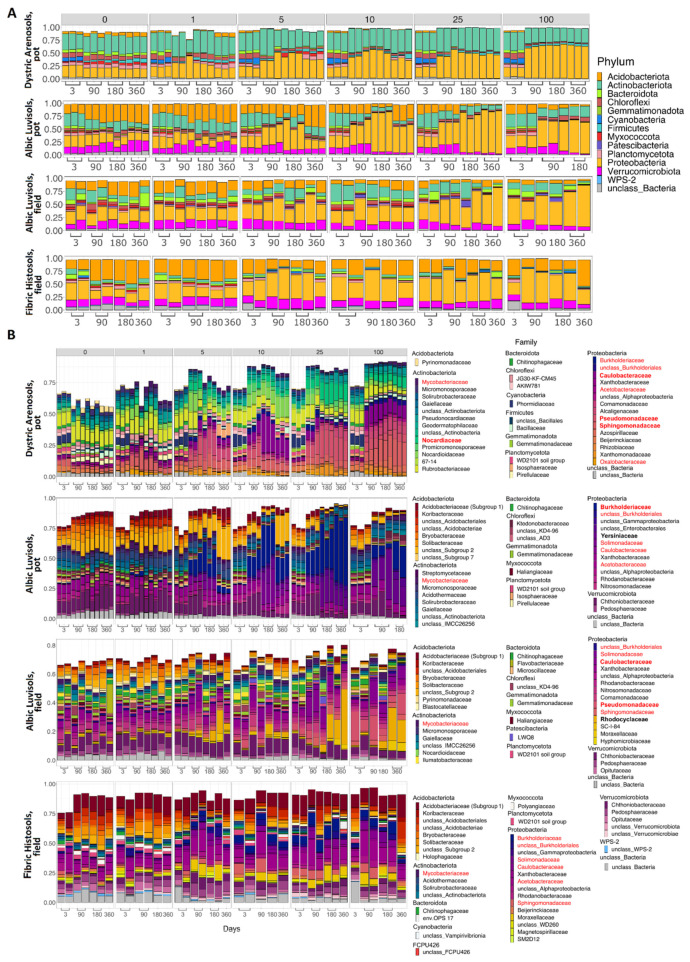
The relative abundance of the top 10 most frequent bacterial phyla (**A**); the top 40 most frequent bacterial families in the studied soils (**B**). Families are sorted and colored according to phyla. The numbers above the histograms represent the initial kerosene loads, in g/kg. The bacterial composition was assessed using the V3V4 region of 16S rRNA; the results for the V4V5 region are provided in Appendix A. The families whose relative abundance was increased after kerosene pollution in all, or almost all, soils are shown in red in the legend. The dominant families whose relative abundance increased after kerosene pollution in a specific soil type and experiment setup are shown in bold.

**Figure 4 life-12-00221-f004:**
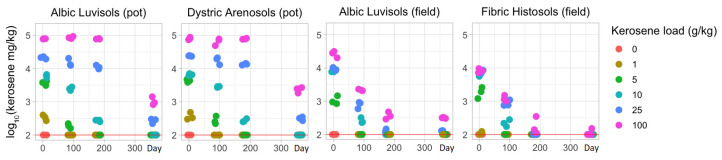
Temporal changes in the concentration of kerosine (g/kg) during the pot and field experiments (0, 1, 5, 10, 25, and 100 are the initial concentrations of kerosene in g/kg). The red line indicates the lowest detectable level.

**Figure 5 life-12-00221-f005:**
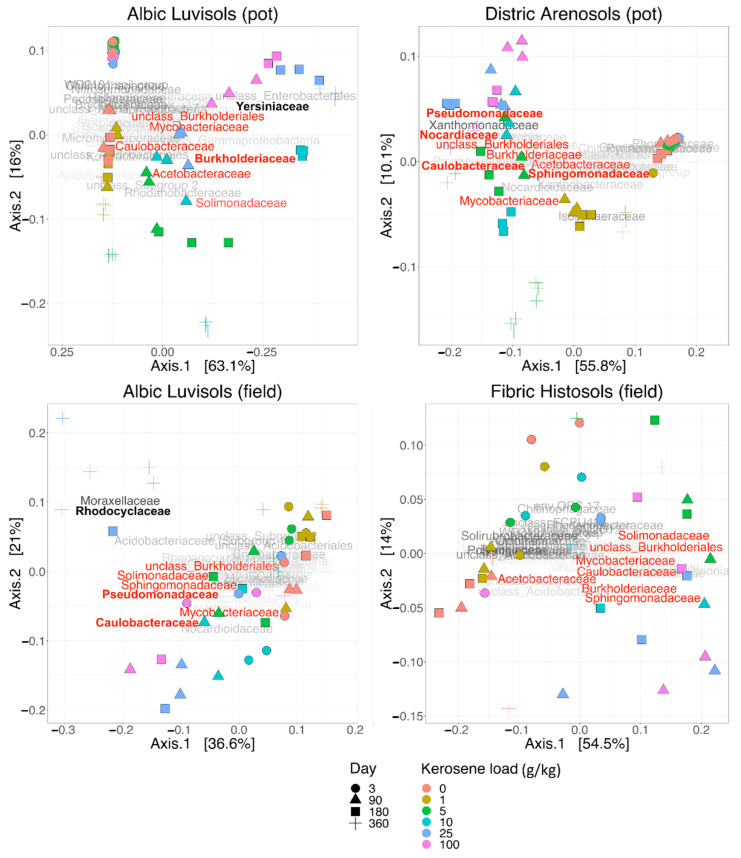
Principal coordinate analysis (PCoA) plots for soils with different initial kerosene loads. The load in g/kg is shown by the dot color, and the day after kerosene treatment is shown by the dot shape (see legend). The beta-diversity is estimated by the weighted UNIFRAC metric (the V3V4 16S rRNA region). The names of the most abundant bacterial families mark the main shifts in the soil microbial composition. The families whose relative abundance increased after kerosene pollution in all, or almost all, soils are in red. Dominant families whose relative abundance increased after kerosene pollution in a specific soil type and experimental setup are in boldface. The PCoA for the V4V5 region and for Bray–Curtis dissimilarity are in Appendix A.

**Figure 6 life-12-00221-f006:**
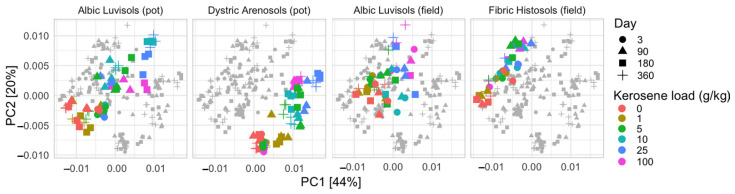
PCA plot based on the relative abundance of metabolic pathways in samples, as predicted by Picrust2 on the V3V4 data. All soils were plotted together in one plane and then separated to four subplots for better readability. Separate PCA plots for individual soils are shown in Appendix A.

**Figure 7 life-12-00221-f007:**
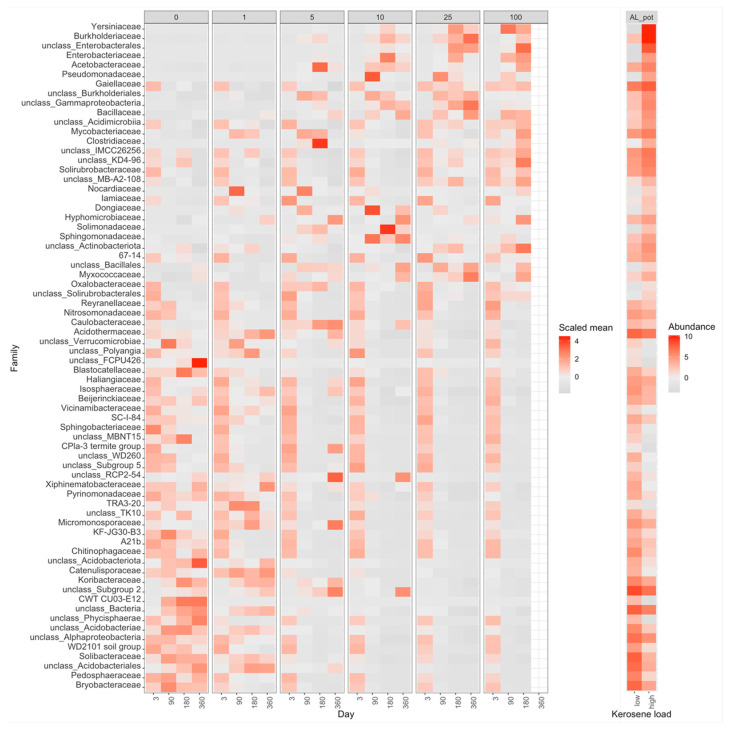
Heatmap with the relative abundance of bacterial families with a significant difference between highly contaminated (kerosene load = 25 or 100, day > 3) and control (kerosene load = 0, day > 3) samples of Albic Luvisols, the pot experiment. In the left panel, the abundance of each bacteria family is scaled (Z-score) across all samples to highlight the relative changes between samples. In the right panel, the medians of-centered log-ratio (clr) transformed abundances of bacterial families in highly contaminated and control samples are shown. The V3V4 data are shown, a similar heatmap for the V4V5 region is shown in Appendix A.

**Figure 8 life-12-00221-f008:**
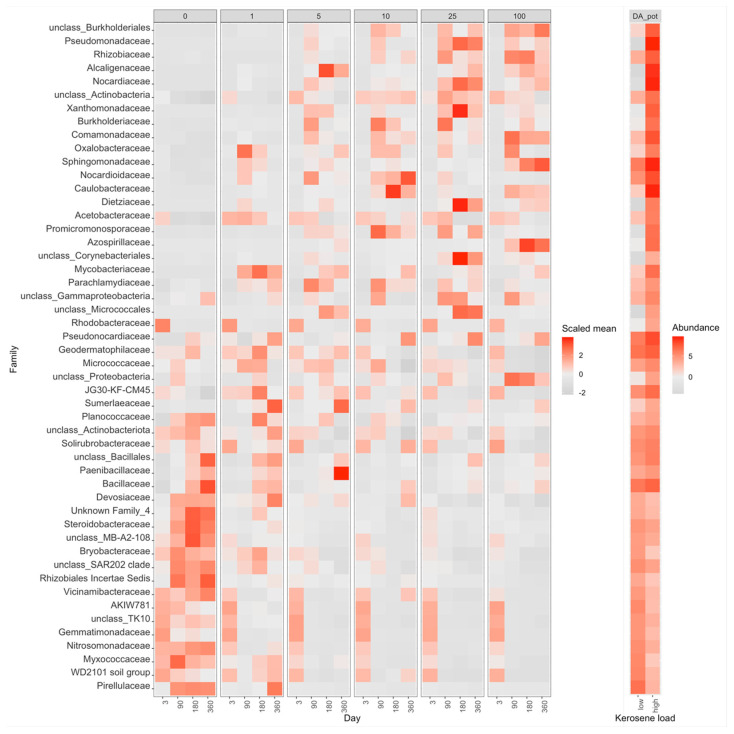
Heatmap with the relative abundance of the top 50 most abundant bacterial families with significant differences between highly contaminated (kerosene load = 25 or 100, day > 3) and control (kerosene load = 0, day > 3) samples of Dystric Arenosols, the pot experiment. Notation as in Figure 7. The V3V4 data are shown. All bacterial families with significant differences between highly contaminated and control samples according to V3V4 data are shown in Appendix A, a similar heatmap for the V4V5 region is shown in Appendix A.

**Figure 9 life-12-00221-f009:**
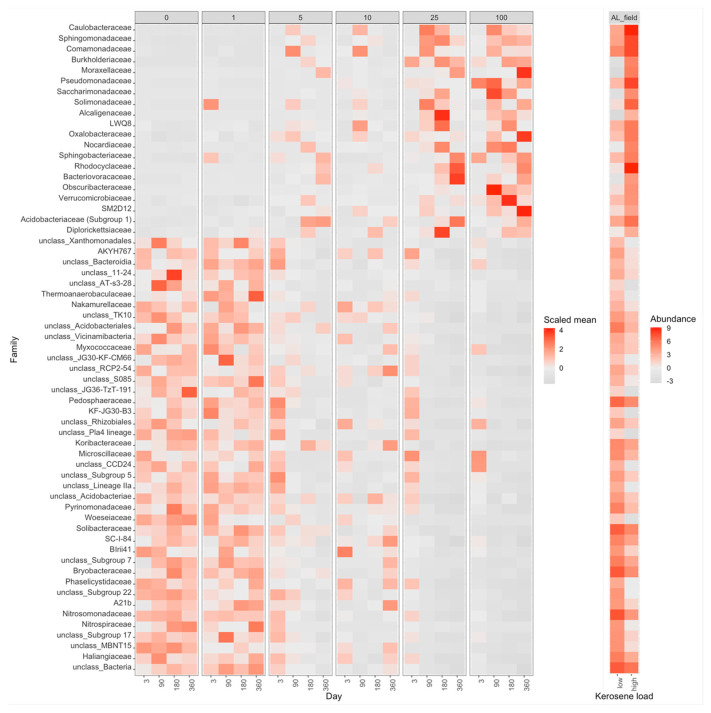
Heatmap with the relative abundance of bacterial families with the significant difference between the highly contaminated (kerosene load = 25 or 100, day > 3) and control (kerosene load = 0, day > 3) samples of Albic Luvisols, the field experiment. Notation as in Figure 7. The V3V4 data are shown, a similar heatmap for the V4V5 region is shown in Appendix A.

**Figure 10 life-12-00221-f010:**
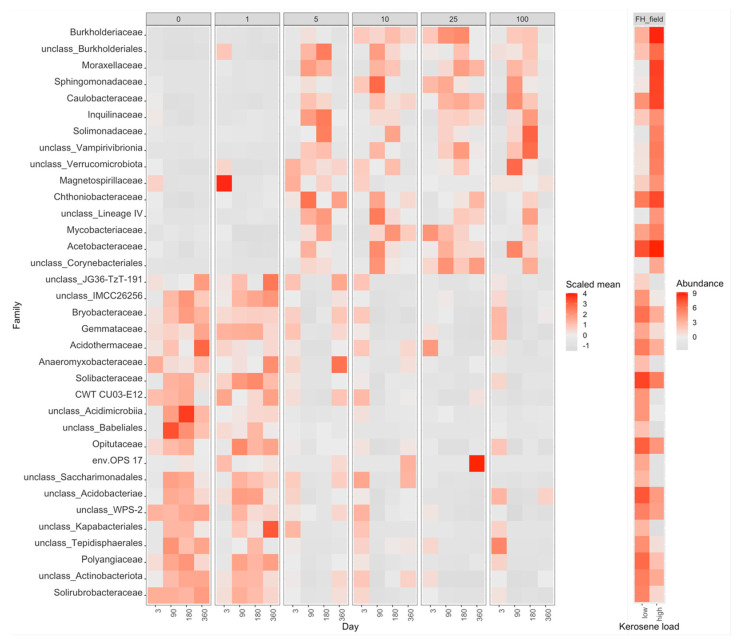
Heatmap with the relative abundance of bacterial families with significant differences between highly contaminated (kerosene load = 25 or 100, day > 3) and control (kerosene load = 0, day > 3) samples of Fibric Histosols, the field experiment. Notation as in Figure 7. The V3V4 data are shown, a similar heatmap for the V4V5 region is shown in Appendix A.

## Data Availability

Raw sequencing reads are deposited at SRA PRJNA786393.

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
