# Peer review of "The Influence of Kerosene on Microbiomes of Diverse Soils"

_life, 2022, doi:10.3390/life12020221_

Round 1

Reviewer 1 Report

The authors can reduce the methodology part to simple version. There are lot of supplementary file or figures. Is there any novelty in this work?. Section 2.4 can be reduced, why detailed version is given?

Author Response

Comments

Response

Extensive editing of English language and style required

The revised version of the manuscript was checked by a professional translator.

The authors can reduce the methodology part to simple version.

We shortened the methodology part to simplify version.

There are lot of supplementary file or figures. Is there any novelty in this work?

Yes. All findings are novel.

Section 2.4 can be reduced, why detailed version is given?

We described the procedure to make it fully reproducible if needed. However, we shortened the section 2.4, as suggested.

Reviewer 2 Report

Dear editor

I write in reference to the manuscript “The influence of kerosene on microbiomes of diverse soils”. The manuscript refers to the evolution of the bacterial microbiota in three soils from the Russian Federation experimentally contaminated with kerosene, employing a microcosmos, and field experiments. Although it is not completely novel, in my opinion, the manuscript is an interesting piece of work that provides useful information of the microbiota response to a common anthropocentric disturbance in different circumstances and can be published after some modifications.

Specific concerns:

Line 45. Please explain what kerosene is, and why kerosene was selected as a model of hydrocarbons contamination.

Lines 53 and 54. Please use cursive to write genus and species.

Lines 54 to 61. Duplicated with lines 43-49.

Lines 80 to 84. “The soils of the Kaluga… an airport is situated”. The lines would be better in the discussion section supporting the selection of the soils used.

Lines 85 to 99. Description of sampling locations would be better in M&M section.

Lines 105 to 110. This paragraph could be moved to the result section or deleted.

Line 105. Please provide the coordinates of the sampling locations

Lines 125 to 135. It is confusing that supplementary tables A1-A5 were cited here (with the table title included). The sentence is referred to subsamples. Probably is an edition mistake. Please delete.

Lines 138 and 139. The argument would be more appropriate in the introduction.

Line 309. (Figure 1). Please explain in the caption why circles and triangles are used in the graphs.

Line 329. (Figure Supplementary figure A 4 ). Please explain the difference between the four graphs in the upper part of Supplementary figure A 4 and the four graphs in the lower part of the same figure. The explanation may be in the figure caption.

Line 345 (Table S3) Are values in the Table S3 means +/- EE? Please clarify.

Line 358. (Figure 4). Y axes must say ln instead log.

Lines 372-373. I could not find SOM in Supplementary Table A 4, A column is titled as "TOC,%" but does not correspond with the text in lines 372-374. Please include SOM in Supp Table A 4 and define TOC,%.

Line 343. I could not identify the two halves of the plot, please clarify (or explain).

Line 453. This is in fact my more important concern. Supplementary Table A 5,  It is not clear to me which are the conditions that are being compared in Supplementary Table A5, the expression "experiments within specified groups" in the table title is specially confusing to me. Please clarify.

Lines 665 to667. Conclusion 3. This conclusion may be right, but is not supported in the results and discussion of the present work.

Author Response

Comments

Response

Line 45. Please explain what kerosene is, and why kerosene was selected as a model of hydrocarbons contamination.

Done.

Lines 53 and 54. Please use cursive to write genus and species.

Corrected as suggested.

Lines 54 to 61. Duplicated with lines 43-49.

Corrected.

Lines 80 to 84. “The soils of the Kaluga… an airport is situated”. The lines would be better in the discussion section supporting the selection of the soils used.

Moved as suggested.

Lines 85 to 99. Description of sampling locations would be better in M&M section.

Moved as suggested.

Lines 105 to 110. This paragraph could be moved to the result section or deleted.

Moved as suggested.

Line 105. Please provide the coordinates of the sampling locations

Done

Lines 125 to 135. It is confusing that supplementary tables A1-A5 were cited here (with the table title included). The sentence is referred to subsamples. Probably is an edition mistake. Please delete.

Corrected as suggested.

Lines 138 and 139. The argument would be more appropriate in the introduction.

Moved as suggested.

Line 309. (Figure 1). Please explain in the caption why circles and triangles are used in the graphs.

This is probably a misunderstanding. The legend was given in the upper part of Figure 1. Now it is given in the lower part of Figure 1

Line 329. (Figure Supplementary figure A 4 ). Please explain the difference between the four graphs in the upper part of Supplementary figure A 4 and the four graphs in the lower part of the same figure. The explanation may be in the figure caption.

On Figure A4 and A6, top panels are based on data on sequencing of V3V4 region and bottom panels are based on V4V5 region. We added the explanation to the revised figure.

Line 345 (Table S3) Are values in the Table S3 means +/- EE? Please clarify.

In supplementary table A3 mean and standard deviation are represented.

Line 358. (Figure 4). Y axes must say ln instead log.

Yes, thank you. In fact, It had been a natural logarithm. We replaced it with the decimal logarithm in revised figure 4.

Lines 372-373. I could not find SOM in Supplementary Table A 4, A column is titled as "TOC,%" but does not correspond with the text in lines 372-374. Please include SOM in Supp Table A 4 and define TOC,%.

We replaced TOC with SOM.

Line 343. I could not identify the two halves of the plot, please clarify (or explain).

We corrected figure 5 and the phrase.

Line 453. This is in fact my more important concern. Supplementary Table A 5,  It is not clear to me which are the conditions that are being compared in Supplementary Table A5, the expression "experiments within specified groups" in the table title is specially confusing to me. Please clarify.

We corrected Supplementary Table A 5.

Lines 665 to667. Conclusion 3. This conclusion may be right, but is not supported in the results and discussion of the present work.

We added the phrase described this conclusion in the R&D section.

Reviewer 3 Report

1. Abstract. All acronyms should be included in this section
2. The authors should provide shorter keywords."
3. The authors should not use quotation marks when mentioning figures, equations, or tables in the text.
4. The presence of microorganisms should be checked with a specific instrument.
5. The discussion of micrographs and infrared spectra is extensive.
6. Authors should not confuse the kerosene oil and other oils
7. Authors should evaluate the microorganism after detection.
08. The results of this work should be compared with those which use kerosene and other oils

Author Response

Comments

Response

1. Abstract. All acronyms should be included in this section

According to Instructions for Authors (https://www.mdpi.com/journal/life/instructions) “Acronyms/Abbreviations/Initialisms should be defined the first time they appear in each of three sections: the abstract; the main text; the first figure or table. When defined for the first time, the acronym/abbreviation/initialism should be added in parentheses after the written-out form.”

As we did not use acronyms in the abstract, we did not explain them there.

2. The authors should provide shorter keywords

According to Instructions for Authors (https://www.mdpi.com/journal/life/instructions) “Three to ten pertinent keywords need to be added after the abstract”. In our manuscript ten keywords are given.

3. The authors should not use quotation marks when mentioning figures, equations, or tables in the text.

We deleted quotation marks when mentioning figures, equations, or tables in the text.

4. The presence of microorganisms should be checked with a specific instrument.

The presence of microorganisms was checked using techniques described in section “2.5. 16S rRNA sequencing data analysis”. This is sufficient by the current standards.

5. The discussion of micrographs and infrared spectra is extensive.

It seems to be some kind of misunderstanding. We did not provide micrographs and infrared spectra in the manuscript.

6. Authors should not confuse the kerosene oil and other oils

Corrected.

7. Authors should evaluate the microorganism after detection.

We aimed at analysis of the total genetic material of the soil and its changes in response to kerosene pollution. We understand that not all bacteria annotated by 16S rRNA are alive at a given time and not all of them are cultivated, but this is not relevant given the stated goals.

08. The results of this work should be compared with those which use kerosene and other oils

Our paper fills gaps in knowledge on the environmental consequences of the contamination with kerosene. The context for the study is given in the introductory section; however, a direct comparison with existing literature is not possible, as there are no similar studies.

Round 2

Reviewer 2 Report

Dear editor. In my opinion, the observations has been appropriately responded and the manuscript can be published.